# Development of Methotrexate Complexes Endowed with New Biological Properties Envisioned for Musculoskeletal Regeneration in Rheumatoid Arthritis Environments

**DOI:** 10.3390/ijms231710054

**Published:** 2022-09-02

**Authors:** Daniel Fernández-Villa, Rosa Ana Ramírez-Jiménez, Inmaculada Aranaz, Niuris Acosta, Blanca Vázquez-Lasa, Luis Rojo

**Affiliations:** 1Instituto de Ciencia y Tecnología de Polímeros (ICTP) CSIC, 28006 Madrid, Spain; 2Centro de Investigación Biomédica en Red de Bioingeniería, Biomateriales y Nanomedicina (CIBER-BBN), 28029 Madrid, Spain; 3Instituto Pluridisciplinar, Departamento de Química en Ciencias Farmacéuticas, Facultad de Farmacia, Universidad Complutense de Madrid, 28040 Madrid, Spain

**Keywords:** methotrexate, strontium, zinc, magnesium, glycosaminoglycans, nitric oxide, anti-inflammatory properties, chitosan, microparticles

## Abstract

Methotrexate (MTX) administration is the gold standard treatment for rheumatoid arthritis (RA), but its effects are limited to preventing the progression of the disease. Therefore, effective regenerative therapies for damaged tissues are still to be developed. In this regard, MTX complexes of general molecular formula M(MTX)·xH_2_O, where M = Sr, Zn, or Mg, were synthesized and physicochemically characterized by TGA, XRD, NMR, ATR–FTIR, and EDAX spectroscopies. Characterization results demonstrated the coordination between the different cations and MTX via two monodentate bonds with the carboxylate groups of MTX. Cation complexation provided MTX with new bioactive properties such as increasing the deposition of glycosaminoglycans (GAGs) and alternative anti-inflammatory capacities, without compromising the immunosuppressant properties of MTX on macrophages. Lastly, these new complexes were loaded into spray-dried chitosan microparticles as a proof of concept that they can be encapsulated and further delivered in situ in RA-affected joints, envisioning them as a suitable alternative to oral MTX therapy.

## 1. Introduction

Rheumatoid arthritis (RA) is one of the most common and prevalent chronic inflammatory diseases worldwide, with its prevalence being around 0.5–1% in the last two decades [1]. This inflammatory environment ultimately leads to articular bone and cartilage erosion, pain, and disability, involving both social and economic burdens [2]. Nowadays, both the European League Against Rheumatism and the American College of Rheumatology recommend the combined use of low-dose methotrexate (MTX), which is a disease-modifying antirheumatic drug (DMARD), and short-term glucocorticoids as the initial treatment [3,4]. The success of these therapies achieves remissions or a targeted low disease profile, preventing joint destruction, disease progression, and even reducing comorbidity risks [5].

There are some reports on the healing of rheumatic erosions in a small proportion of patients undergoing MTX therapy, although these events are not common [6,7]. It is believed that this regenerative process mainly takes place in small joints, although there are some cases reported in larger ones, like the hip joint [6,7]. However, there are multiple options to try to improve this physiological regenerative process.

In this regard, the use of bioactive ions is of particular interest because of their high tunability, stability, reduced cost, and non-immunogenicity [8]. For instance, strontium is a well-known inductor of mesenchymal stem cells (MSCs) differentiation towards the osteogenic lineage, as well as it promotes their proliferation and inhibits the resorptive activity of osteoclasts [9,10,11]. In addition, it has been shown to increase glycosaminoglycans (GAGs) deposition by chondrocytes in vitro [12]. Similarly, zinc plays a role in musculoskeletal development and cell division processes [8]. It has been demonstrated to increase the mineralization degree of the extracellular matrix and the collagen deposition by MSCs and osteoblast-like MC3T3-E1 cells, respectively. It also antagonizes osteoclastogenesis, promoting osteoblastic differentiation and increasing the bone formation rate in a dose-dependent manner [9]. Moreover, some anti-inflammatory properties have been described [13]. Finally, magnesium has been shown to promote chondrogenic differentiation of MSCs and inhibit macrophage-induced inflammation [14]. It is also essential for the DNA and RNA synthesis processes and has been found to be significantly reduced in the serum of RA patients [15]. For all these reasons, these cations, among others, have been extensively incorporated into multiple biomaterials envisioned for regenerative applications [16,17,18]. In this regard, our research group has prepared metallic complexes of folic acid with Ca, Sr, Zn, Mg, and Mn, and studied them in the context of musculoskeletal regeneration. As they were shown to upregulate genes related to osteo- and chondrogenesis, as well as to promote physiological processes which are essential during the regeneration of these tissues, they were incorporated into different scaffolds, improving their performance both in vitro and in vivo [10,11,19,20].

In this framework, there are some published papers reporting the synthesis and physicochemical characterization of complexes of MTX with bioactive cations such as Sr, Zn, Cu, or Pt [21,22,23,24]. Exceptionally, in two of these works the authors addressed some anti-proliferation studies and loaded them into a controlled delivery system [23,24]. Consequently, the additional positive effects of MTX complexes coming from the bioactive cation have not been explored yet and research on this topic is required prior to their final biomedical application.

Another aspect to consider is that, in the treatment of RA, MTX is commonly administered orally. After the ingestion, MTX is rapidly absorbed in the small intestine, reaches its maximum concentration in plasma within 1–2 h, and becomes undetectable at 24 h due to its urinary excretion [25]. Despite having the safest profile among all the conventional synthetic DMARDs used for RA, this systemic administration sometimes results in adverse side effects, ranging from non-healing ulcers to lymphoma-like lesions [26,27,28]. This fact, in combination with MTX pharmacokinetics, makes its encapsulation highly interesting. In this sense, some efforts have been made to load it into systems of various natures with different applications [29]. For example, Zhao et al. aimed for a therapeutic effect and they loaded MTX into poly(lactic-co-glycolic acid)-PK3 (a polyketal) nanoparticles functionalized on their surface to promote their uptake by activated macrophages and they almost restored the normal phenotype on an adjuvant-induced arthritis rat model [30]. In contrast, Nogueira et al. achieved a prophylactic effect when administering a liposomal formulation containing MTX to an arthritic mice model before the disease onset, an effect which was not obtained with the soluble MTX drug [31]. Thus, these strategies direct the drug towards the affected tissues, avoiding systemic toxicities and administering minimal doses.

In this work, we have developed and characterized three different MTX complexes bearing bioactive cations (i.e., Sr, Zn, and Mg) and studied the synergic combined effect of each of the individualized moieties. Indeed, this is the first time that a MTX derivative is shown to not only maintain the immunosuppressing effects of MTX, but also to promote biological processes related to cartilage regeneration in vitro. Furthermore, the compounds described in this work have been loaded into spray-dried chitosan microparticles as a proof of concept that they can be encapsulated with a biocompatible polymer for an ulterior controlled release envisioned for an in situ administration in RA-affected joints.

## 2. Results

### 2.1. Synthesis of MTX Derivatives

Complexes of MTX with strontium, zinc, and magnesium were synthesized as described in Figure 1 and purified in high yields (>95%).

The synthesis process retrieved highly pure products, as shown in Figure 2, by Energy Dispersive X-ray analysis (EDAX). MTX derivatives were isolated with presence of minor chlorine impurities, always under 1% in weight.

### 2.2. Physicochemical Characterization of MTX Derivatives

Molecular formulas and their corresponding molecular weights for the resulting MTX derivatives were calculated by determining the content of complexed metals by inductively coupled plasma (ICP) spectroscopy, and for the coordinated water molecules by thermogravimetric analysis (TGA). Results are shown in Table 1.

Thermograms obtained by TGA are shown in Figure 3a. In all the cases, there was a first weight loss up to 100–120 °C, which corresponded to the coordinated water molecules within the complexes. Above 120 °C, two types of degradation behavior were observed depending on the compound. On the one hand, curves of MTX and SrMTX displayed a plateau until 237 °C and 305 °C, respectively, and over these temperatures the maximum degradation process started, diminishing high mass percentages in short temperature increments. In contrast, there was no plateau in the curves of ZnMTX and MgMTX, but an uninterrupted loss of mass until 300 °C and 303 °C, respectively, when the compounds reached their maximum degradation rate. First derivative curves (DTG) for MTX and its metallic derivatives are shown in Figure 3b, where the first and second peaks represent the maximum temperature of dehydration and degradation, respectively. These results are summarized in Appendix A.

Regarding X-ray diffraction (XRD) analysis, the main reflections of MTX and its derivatives with their corresponding 2θ and d-spacing values were determined from their diffractograms (Figure 3c). While MTX and SrMTX showed well-resolved diffractograms, ZnMTX and MgMTX retrieved broader peaks.

To fully characterize the interaction between MTX and the different cations, attenuated total reflectance–Fourier-transform infrared spectroscopy (ATR–FTIR) analyses were performed. The spectra are shown in Figure 4, while the main vibrational bands are summarized in Appendix A.

The most relevant bands corresponded to vibrational frequencies of the carboxylic group, which ranged between 1800 and 1400 cm^−1^. The band at 1682 cm^−1^, corresponding to the vibrational frequency of the free carboxylic group (COOH), was not present in the spectra of the MTX complexes. In addition, symmetric and asymmetric vibrational frequencies of carboxylate groups of MTX derivatives shifted to lower and higher wavenumbers respectively when compared with non-complexed MTX. The bands from 1206 to 1356 and 765 cm^−1^, which corresponded to C-N stretching absorptions of aromatic amines and out-of-plane wagging of primary amines, respectively, maintained their relative intensities after complexation.

NMR analyses were also conducted to identify possible additional binding sites. Two-dimensional spectra, ^13^C assignations, and chemical shifts are shown in Appendix A. Chemical shifts from the carbons located near to the carboxyl groups in MTX significantly varied when compared to the derivatives, while the p-aminobenzoic portion of the molecule was not affected by the complexation. Regarding the diamino-pteridine ring, only C13 and C15 could be assigned due to the low solubility of the drugs, which varied too.

### 2.3. In Vitro Biological Evaluation

#### 2.3.1. Cell Viability Studies

Confluent cultures of Human Chondrocytes-articular (HC-a), human fibroblast-like synoviocytes (Osteoarthritis phenotype, OA-HFLS), and RAW264.7 murine macrophages were treated with different doses of MTX and its derivatives, and their cytotoxic effects were evaluated by monitoring changes in mitochondrial activity by the alamarBlue^®^ method. As shown in Figure 5, MTX derivatives showed an analogous cytotoxic profile when compared with MTX in all the cases. Values of half maximal inhibitory (IC50) and effective (EC50) concentrations for each compound and cell line, calculated by non-linear regression analyses, are included in the Appendix A.

Regarding OA-HFLS cells, none of the analyzed compounds diminished the mitochondrial activity of this cell type below 70% along their range of solubility. Similarly, MTX, SrMTX, and MgMTX did not have cytotoxic effects when applied to cultures of HC-a cells. In contrast, ZnMTX showed a dual behavior. Doses of ZnMTX below 30 μM were shown to stimulate the mitochondrial activity of HC-a cells, whereas doses over 80 μM were found to be lethal for this cell type (Figure 5c).

On the other hand, doses of MTX and SrMTX ranging between 0.7 and 1100 μM diminished 60% of the mitochondrial activity of RAW264.7 cultures. Similar results were obtained for MgMTX, but to a lesser extent, reaching a 50% value at 550 μM. In contrast, ZnMTX followed the same cytotoxic profile as MTX below 230 μM, while being lethal over this concentration.

#### 2.3.2. Effect on the Deposition of GAGs by Chondrocytes

To evaluate the regenerative potential of these newly synthesized MTX derivatives, cultures of HC-a cells were treated with cytocompatible doses of each compound, and the contents of GAGs deposited in the matrix after 14 days of culture were quantified.

As shown in Figure 6, treating HC-a cells with MTX or its derivatives enhanced the production of GAGs per cell at both assayed concentrations compared to untreated control cultures. In addition, increasing the concentration of MTX from 30 to 140 μM significantly diminished this indicator of new matrix formation. On the other hand, regarding the synthesized derivatives, 30 μM ZnMTX doses did not alter GAGs production if compared to the condition treated with 30 μM MTX but increased it when compared to 140 μM MTX-treated cultures. Furthermore, doses of 140 μM Sr- and MgMTX increased GAGs deposition compared to the same dose of MTX.

#### 2.3.3. Effect on the Production of Nitric Oxide (NO) by LPS-Stimulated Macrophages

Additional anti-inflammatory effects of MTX derivatives were tested by measuring the NO released by cultures of LPS-stimulated macrophages. Results of the quantification of released NO normalized against cell number are shown in Figure 7a, while the mitochondrial metabolic activity values of analogue cultures of macrophages, but non-stimulated with LPS, are plotted in Figure 7b.

As shown in Figure 7a, neither MTX nor SrMTX reduced the release of NO by RAW264.7 macrophages at any assayed concentration. Only the dose of 140 μM ZnMTX was shown to decrease the release of NO by this cell type, while doses of 225 μM ZnMTX and MgMTX increased this parameter.

### 2.4. Encapsulation of MTX Derivatives into Chitosan Microparticles

#### 2.4.1. Characterization of Chitosan Microparticles

Chitosan microparticles loaded with the various MTX derivatives were synthesized by spray-drying and further characterized (Table 2).

Yields of production were higher for the microparticles loaded with MTX derivatives compared to the commercial drug under the chosen conditions. Size dispersion was evaluated by Coulter analysis, obtaining monodisperse populations of around 5- to 10-μm diameter for all the samples, except for ZnMTX-loaded ones, where a second population was found around 20–30 μm (Figure 8a), being larger in terms of the mean diameter and presenting a higher dispersion as assessed by the span value (Table 2), which might be attributed to microparticles aggregation. The spherical morphology of all the resulting microparticles was later confirmed by SEM, as shown in Figure 8b. However, while MTX-loaded microparticles displayed a smooth surface, the ones loaded with the metallic MTX derivatives presented rougher surfaces. Lastly, encapsulation efficiencies for each compound were considerably high, as shown in Table 2, especially for SrMTX and MgMTX, which were over 90%. It must be noted that a final concentration of 4% (*w*/*w*) ZnMTX was selected, in contrast to the 8% (*w*/*w*) loaded for the rest of the compounds, as over that concentration zinc ions led to chitosan precipitation.

#### 2.4.2. In Vitro Drug Release

Drug release kinetics from the loaded microparticles were analyzed in a medium-term release assay (Figure 8c). No burst effects were observed in any loaded system but a progressive release with time in all the cases. Half-life values were 68.14, 78.86, 87.87, and 98.62 min for MTX, SrMTX, ZnMTX, and MgMTX, respectively. Complete drug release of all the compounds was reached in 24 h.

## 3. Discussion

While being established as the gold standard treatment for RA, MTX administration has still some drawbacks such as the limited tissue regeneration of the inflamed joints after the treatment and the secondary adverse effects derived from its prolonged oral uptake [26,32]. To fulfil these necessities, we proposed to complex MTX with different bioactive divalent cations to combine the individual positive effects of each moiety, and to load them into a common delivery system that serves as a proof of concept that these complexes can be encapsulated for an ulterior controlled release with in situ application perspectives.

In this work, we have addressed the first aim by synthesizing and characterizing MTX complexes bearing cations of strontium, zinc, and magnesium, and by demonstrating that the complexation of these metals adds new features of interest to the marketed drug. Prior to this work, there were only a few articles about metal-based derivatives of MTX. Two of them just focused on the physicochemical characterization of the reported complexes (with strontium and zinc) [21,22], while in two other works the authors first reported some anti-proliferation studies as well as the characterization of the complexes (bearing zinc, copper, and platinum), and then they employed the complexes as model drugs to be released from alginate-carboxymethyl cellulose composite beads [23,24]. Thus, this is the first time that metal-containing MTX derivatives are studied in the context of an autoimmune disease, seeking pro-regenerative effects, and this work encompasses the most in-depth biological characterization carried out so far with this kind of complexes.

### 3.1. Synthesis and Physicochemical Characterization of MTX Derivatives

The complexes described here presented a different composition and/or structure when compared to the ones already reported in the literature, which is probably due to the differences in the synthesis procedures. For example, a hydrated strontium salt of MTX was reported in the work of Mastropaolo et al., but the crystals were grown from a solution of ethanol/water at pH 7.5 and at 5 °C by a very slow evaporation method, leading to a salt whose asymmetric unit was composed of two MTX molecules coordinating two strontium ions [21]. In contrast, the method selected in our work considered both the heating at 50 °C and an aqueous reaction medium to ensure the formation of the complexes in solution and not in suspension (Figure 1). This procedure was previously optimized by our group to synthesize folic acid derivatives containing divalent cations [10,19], and has also been successful now for obtaining the analogue MTX ones, retrieving yields over 95% in all the cases, with minimum chlorine impurities and no other co-precipitates, as detected by EDAX (Figure 2). This procedure, in combination with TGA and ICP results, allowed us to propose the general molecular formulas of these complexes, being MC_20_H_22_N_8_O_5_·xH_2_O, where M = Sr, Zn, or Mg, and X = 4, 1, and 5, respectively. It is worth highlighting that the presence of persistent metal chlorine impurities after purification and the instrument-associated error may be responsible for the discrepancies between calculated and found values for metal contents in the complexes [33].

On the other hand, the procedures carried out by Nodiţi et al. and Çeşme et al. to synthesize complexes of MTX with Zn^2+^ were more similar to the one followed here, except for the ratio of ZnCl_2_ to MTX that was 1:2 and 1:1, respectively, while an excess of ZnCl_2_ (3:1) was added in our case. These procedures resulted in complexes where one zinc ion was coordinated by two and one MTX molecules, respectively [22,23].

In any case, in all these previous reports, coordination between the cations and MTX occurred exclusively over the free carboxyl groups of MTX as demonstrated by crystallographic analysis, FTIR, and/or NMR spectroscopies [10,19,21,22,23]. In our case, the analysis of the ATR–FTIR spectra revealed, as expected, that the band corresponding to υ(COOH) vibration in the MTX spectrum, located at 1682 cm^−1^, did not appear in the spectra of the MTX derivatives, meaning that the metals were coordinated over the free carboxyl groups of MTX in all the cases (Figure 4). Moreover, the finding that the bands located from 1206 to 1356 and 765 cm^−1^, which corresponded to C-N stretching absorptions of aromatic amines and out-of-plane wagging of primary amines, respectively, appear in the same spectral range after the complexation, suggests that amine groups are not involved in the formation of additional dative bonds [22].

Nevertheless, NMR analyses were also conducted to corroborate the absence of additional binding sites. Two-dimensional spectra, ^13^C assignations, and chemical shifts are shown in Appendix A. Similar to the FTIR analysis, chemical shifts from the carbons located in the proximities of the carboxyl groups in MTX significantly varied when compared to the derivatives, while the *p*-aminobenzoic part of the molecule was not affected by the complexation. Regarding the diamino-pteridine ring, although only C13 and C15 could be assigned, due to the low solubility of the drugs, their chemical shifts changed too. We hypothesize that these variations might be attributed to electronic rearrangements within the pteridine ring after the molecule loses its zwitterionic state due to the complexation over the carboxyl group and to intermolecular interactions. As previously studied, cationic and zwitterionic (i.e., M-MTX) species displayed average interaction energies twice and thrice higher than anionic ones (i.e., MTX), leading to more energetically favorable states for the formation of clusters (i.e., increased intermolecular interactions) [34].

This coordination over the carboxyl group can have different bonding modes as previously reviewed by Nakamoto [35]. As the COOH groups are expected to be deprotonated under the applied conditions, the carboxylate groups should adopt a C_2v_ symmetry, resulting in symmetric (υ_s_(COO^−^)) and asymmetric vibrations (υ_asy_ (COO^−^)), and their shifts to lower or higher frequencies reveal the binding mode that they exhibit. In our case, the shift of symmetric and asymmetric (COO^-^) vibrations of all the complexes to lower and higher frequencies respectively indicated a monodentate binding geometry (Appendix A). Interestingly, the Zn^2+^-containing complex reported by Çeşme et al. also presented a 1:1 stoichiometry, but the coordination occurred via a bidentate binding geometry [23]. Thus, these are the first MTX complexes reported with a stoichiometry 1:1 (Metal:MTX) and a monodentate coordination mode.

Regarding the thermal characterization, the spectra obtained for MTX were similar to that reported by Nodiţi et al. [22]. Three water molecules were released from the MTX structure until 120 °C, as assessed by TGA (Figure 3a), meaning that MTX was thermally unstable as a trihydrate. Similarly, MTX derivatives were also unstable in their hydrated form, corresponding the first weight loss step to a dehydration phase. However, Zn- and MgMTX presented some differences in this first step compared to MTX and SrMTX. While the trace of the latter had a plateau after this phase, meaning that their anhydrous form was stable until the onset of the second degradation step, the former continuously degraded along that range of temperatures. This could be attributed to the lower degree of crystallinity that Zn- and MgMTX present, as shown in their corresponding diffractograms by their broader peaks (Figure 3c), accounting for that first step to the degradation of the amorphous phase of the drugs. Conversely, MTX and SrMTX showed well-resolved diffractograms with defined peaks corresponding to tetragonal and primitive triclinic crystal structures respectively, as previously reported [21,36]. Thus, the complexation of the divalent cations increased the thermal stability of MTX in all the cases, increasing the temperature for a 50% of weight loss (T_50%_) from 332 °C to 476 °C, >650 °C, and 432 °C, and the remanent mass at 600 °C from 34.39% to 43.87%, 56.14%, and 38.42% for Sr-, Zn-, and MgMTX, respectively.

### 3.2. In Vitro Biological Evaluation of MTX Derivatives

The study of the biological activity of the different MTX derivatives compared to the marketed drug is crucial to assess their feasibility for use as a treatment for RA. In this sense, we evaluated their cytotoxic effects on different cell lines related to the tissues that make up the joints, their effects on GAGs deposition, and additional anti-inflammatory properties.

*Cell viability studies*. Although MTX is the first-line drug for the treatment of RA, its mechanism of action is still not fully understood. It is well established that MTX exerts different biological effects depending on the cell type. For example, while MTX inhibits metalloproteinases production in fibroblast-like synoviocytes, it also induces apoptosis in monocytes and macrophages [32]. In Figure 5, we have addressed this last issue and we have shown that MTX and the synthesized derivatives diminished the cell viability of RAW264.7 murine macrophages while not affecting the viability of primary cultures of human chondrocytes and fibroblast-like synoviocytes. This macrophage-directed cytotoxic behaviour is of particular interest in autoimmune inflammatory diseases such as RA or psoriasis. ANOVA tests for the values of EC50 of MTX and its derivatives showed that there was no statistical difference among the marketed drug and the complexes, demonstrating that the addition of cations to form complexes does not alter the immunosuppressant activity of MTX (Appendix A).

Similarly, IC50 values of MTX, Sr-, and ZnMTX were also non-statistically different. However, MgMTX did not reach an IC50 value along its solubility range of concentrations. This observation can be due to the effects of magnesium which, being essential for DNA replication, has been shown to increase the metabolic activity of macrophages [37,38]. Finally, a distinctive behaviour was observed for the ZnMTX derivative, whose application on macrophage and chondrocyte cultures was lethal at over 200 and 80 μM respectively. This has been attributed to the toxicity of Zn^2+^ ions as it has the same tendency observed by our group in cultures of human bone marrow-derived MSCs treated with ZnCl_2_ and a folic acid derivative bearing a Zn^2+^ ion [19].

*Effect on GAGs deposition by chondrocytes*. GAGs are one of the main components of the extracellular matrix of the joints, serving as a natural lubricant to minimize friction between bones. Moreover, they have also been shown to regulate cell growth and proliferation, promote cell adhesion, and have anticoagulant and wound repair properties. Thus, their loss in the context of RA has several consequences both at macroscopical and cellular levels. Hence, promoting GAGs deposition in the arthritic joint is highly sought to restore the damaged cartilage.

Considering the viability studies and the ranges of bioactivity of the cations, we selected cytocompatible concentrations of the derivatives to test whether the complexation of bioactive cations could increase the deposition of GAGs by chondrocytes. ZnMTX was assayed at a lower concentration (30 μM) than the other complexes (140 μM) as it was found to be cytotoxic for chondrocytes at higher concentrations (IC50 = 66.6 μM).

The ratio between deposited GAGs and the content of DNA is shown in Figure 6. Higher values were obtained for MTX and M-MTX-treated cultures in comparison to control ones due to the cytostatic effect of MTX, which impaired cell proliferation to a higher extent than GAGs deposition. In addition, it was found that higher doses of MTX significantly diminished GAGs deposition, highlighting the importance to enhance this deposition under MTX treatment. In this sense, significant differences were observed for cultures treated with 140 μM SrMTX and MgMTX. This agrees with previous reports showing that Sr promotes chondrogenic processes and stimulates human cartilage matrix formation in vitro [39], and with the Mg’s protective effect mentioned above. In contrast, despite the well-established role of Zn on matrix remodelling, no changes in GAGs deposition were observed at that concentration, which might be attributed to its broad range of bioactivity [8]. Nevertheless, additional assays could be performed to check whether they can stimulate bone regeneration too at these doses.

*Effect on NO production by activated macrophages*. In RA, the NO secreted by macrophages and synoviocytes plays a role in inflammation, angiogenesis, and tissue destruction [40]. Thus, we examined whether the new MTX derivatives diminished this NO release in cultures of LPS-stimulated RAW264.7 macrophages (Figure 7). The only reduction in NO production was achieved for the cultures treated with 140 μM ZnMTX, which has been attributed to the anti-inflammatory properties reported for the Zn^2+^ ion. Indeed, it has been shown that zinc supplementation from 100 to 150 μM significantly reduced the NO production in LPS-stimulated rats in vivo [13]. On the other hand, neither MTX nor the other two derivatives diminished NO release at the assayed concentrations. Thus, new anti-inflammatory properties were only given to MTX by its complexation with Zn^2+^ cations.

### 3.3. Encapsulation of MTX Derivatives into Chitosan Microparticles

Both MTX and the new derivatives have been shown to have potent biological activities, whose effects must be well-controlled in a spatiotemporal manner to avoid side effects. In this sense, in a first approach, we loaded them into microparticles as a proof of concept that they can be encapsulated for a local and controlled release in the damaged joint.

Chitosan was selected to prepare the microparticles because of its well-known biocompatibility, biodegradability, and versatility. Indeed, it has been widely used for the preparation of multiple micro- and nano-sized drug delivery systems [41,42,43]. In this work, microparticles were prepared by spray-drying, which is a common methodology employed in the fabrication of microsystems since it combines rapidity and little dependency on the solubility characteristics of the drug and polymer, achieving microencapsulation in a one-step process which is easy to scale up [44,45]. This is of particular interest for MTX encapsulation due to its reduced solubility in aqueous media (<1 mg/mL).

Little differences were observed among the various loaded microparticles in terms of physicochemical characterization (Figure 8). However, the ones with larger differences were the ones encapsulating ZnMTX, which was reflected, for instance, in their mean size and the final loaded concentration. The encapsulation efficiency was affected by the presence of ZnMTX too, obtaining excellent values over 90% for Sr- and MgMTX-loaded particles and more moderate results for ZnMTX, which were similar to the ones obtained for the commercial drug. Previous studies using other materials and procedures reported similar values [46,47,48].

Finally, regarding the drug release studies, all the microparticles displayed the same tendency when immersed in phosphate-buffered solution (PBS) at 37 °C. Mean half-life values for the systems indicate that cations complexation did not influence the release from the microparticles. In addition, the complete drug release was achieved after 24 h, which is in agreement with some other published data [41,47,48]. Optimizing the times for the delivery of these compounds is beyond the scope of this article, but could be done by modulating physicochemical parameters of the chosen polymer (i.e., molecular weight or chemical modifications of chitosan) or by designing more complex delivery systems (e.g., releasing these particles from hydrogels) [49,50]. However, this serves as a proof of concept that, after achieving the encapsulation of the new derivatives, they can be delivered progressively and with no burst effects.

## 4. Materials and Methods

### 4.1. Materials

MTX was purchased from Glentham Life Sciences (London, UK), strontium chloride hexahydrate (99%) from Acros Organics (Geel, Belgium), zinc chloride (99%) from Fluka Analytical, magnesium chloride hexahydrate (99%) from Merck, Quant-iT™ PicoGreen™ double stranded-DNA (dsDNA) assay kit from Invitrogen, alamarBlue^®^ from Bio-Rad, and Griess reagent (modified), 1,9-dimethyl-methylene blue zinc, and tablets of PBS pH 7.4 from Sigma. Chitosan employed for encapsulation studies (deacetylation degree: 91%, intrinsic viscosity: 2.45 dL/g, M_w_ calculated by viscosity: 47 kDa) was kindly donated by InFiQuS S.L.

### 4.2. MTX Derivatives Synthesis and Characterization

*Synthesis of Metal-MTX complexes*. Three different MTX complexes bearing strontium, zinc, and magnesium (referred here as SrMTX, ZnMTX, and MgMTX, respectively) were synthesized following a previously described analogous methodology for the synthesis of folate derivatives [10,19]. For synthesizing SrMTX, 12 mL of an MTX aqueous solution (0.4 M, 4.8 mmol) was prepared, its pH was adjusted to 7.4 using NaOH 1 M, and it was refluxed at 50 °C for 2 h. Next, 12 mL of a SrCl_2_·6H_2_O solution (1.2 M, 14.4 mmol) in ethanol–water (50% *v/v*) was added to the previous solution and the reaction mixture was further stirred and heated at 50 °C for 1 h. The same protocol was followed for the synthesis of ZnMTX and MgMTX, except that ZnCl_2_·6H_2_O and MgCl_2_·6H_2_O salts were used instead. After this, the reaction was quenched by cooling it down in an ice bath and the newly formed precipitate was collected by filtration. Finally, the dark orange solid product was recrystallized twice with water/ethanol solutions, milled, and dried under vacuum at 50 °C for 72 h until obtaining fine orange crystals. A scheme of the reaction conditions is shown in Figure 1.

*Physicochemical characterization*. The content of strontium, zinc, and magnesium within the molecular formula of complexes was determined by emission spectroscopy analysis using an ICP optical emission spectrometer (Perkin-Elmer 430DV). Briefly, 12.5 mg of each complex was dissolved in 25 mL of hydrochloric acid (HCl) 0.1 M to completely release the cations from the derivatives. Metal concentrations were determined from a calibration curve constructed from the absorbance of a solution series prepared from standard stock solutions of the respective Sr, Zn, or Mg chloride salts in HCl (2% *w/v*). The presence of the cations after the syntheses was evaluated by EDAX performed on a Hitachi SU8000.

XRD of the samples was carried out to analyze their crystalline composition. Analyses were performed on a Bruker D8 Advance instrument which works with CuKα radiation (nλ = 1.542 Å) at a 0.02 step size and at 0.5 s per step. D-spacing corresponding to each crystal was calculated by using the Bragg law equation (nλ = 2d · sin (φ)). To analyze the chemical composition and the coordination modes of MTX derivatives, Fourier-transform infrared (FTIR) spectroscopy was performed on a PerkinElmer Spectrum Two spectrophotometer with an attenuated total reflectance (ATR) attachment. Spectra were smoothed, corrected to ATR, and normalized for drawing conclusions.

The structures of MTX and its derivatives were further elucidated by Nuclear Magnetic Resonance spectroscopy (NMR). NMR spectra were recorded at 318 K, using D_2_O as solvent, on an Agilent SYSTEM 500 NMR spectrometer (^1^H 500 MHz, ^13^C 125 MHz) equipped with a 5-mm HCN cold probe. Chemical shifts of ^1^H (δH) and ^13^C (δC) in parts per million were determined relative to internal standards of solvent residual peak (HDO) in D_2_O (δH 0.00) and external 1,4-dioxane (δC 67.40) in D_2_O, respectively. One-dimensional (1D) NMR experiments (^1^H and ^13^C{^1^H}) were performed using standard pulse sequences. Two-dimensional (2D) [^1^H, ^1^H] NMR experiments [gradient correlation spectroscopy (gCOSY)] were carried out with the following parameters: delay time of 1 s, spectral width of 4432.6 Hz in both dimensions, 32 transients for each of 256-time increments. The data were zero-filled to 1024  × 1024 real points. Two-dimensional (2D) [^1^H−^13^C] NMR experiments [gradient heteronuclear single-quantum coherence (gHSQC) and gradient heteronuclear multiple bond correlation (gHMBC)] used the same ^1^H spectral window, a ^13^C spectral window of 25,133.5 Hz for gHSQC and 30,165.9 Hz for gHMBC, 1 s of relaxation delay, 1024 data points, and 32 transients for each of 256- or 400-time increments. The data were zero-filled to 1024 × 1024 real points.

Thermal degradation of the complexes was assessed by TGA using a thermogravimetric analyzer TGA Q500 apparatus (TA instruments) under dynamic N_2_ at a heating rate of 10 °C/min from 40 to 600 °C.

### 4.3. In Vitro Biological Evaluation

*Cell culture*. In vitro biological tests were performed in three different cell lines. RAW 264.7 murine macrophages were purchased from the European Collection of Authenticated Cell Cultures (ECACC) and cultured in tissue culture T-75 flasks (Sarstedt) with Dulbecco’s Modified Eagle’s Medium (DMEM)-high glucose enriched with 4500 mg/L glucose, sodium pyruvate, and sodium bicarbonate (Sigma-Aldrich) supplemented with 10 vol% of fetal bovine serum (FBS), 2 vol% L-glutamine, and 1 vol% of a solution of penicillin/streptomycin (P/S solution). Primary cultures of Human Chondrocytes-articular (HC-a cells) were purchased from Innoprot (P10970, Batch 6944) and cultured in tissue culture T-75 flasks with a Cell+ growth surface for sensitive adherent cells (Sarstedt) with chondrocyte medium (Innoprot) supplemented with 5 vol% of FBS, 1 vol% of Chondrocyte Growth Supplement, and 1 vol% of P/S solution. Human Fibroblast-Like Synoviocytes: Osteoarthritis (HFLS-OA cells) were purchased from Cell Applications (408OA-05a, Lot. 1776) and cultured in tissue culture T-75 flasks with ready to use HFLS Growth Medium (Cell Applications). All cultures were maintained in a series 800 DH incubator (Thermo Scientific, Waltham, MA, USA) under a 5 vol% CO_2_ atmosphere at 37 °C.

*Cell viability studies*. Cell viability studies were performed on the three aforementioned cell lines. For this, cells were seeded in sterile 96-well culture plates (Thermo Scientific) at a density of 10^5^ cells/mL and cultured for 24 h to allow cell attachment. Next, cells were incubated for 24 h with different concentrations of MTX derivatives (n = 8). Finally, media were removed and 100 μL of an alamarBlue^®^ solution (10% *v/v*) prepared in phenol red-free DMEM—low glucose (Sigma-Aldrich) was added to each well. Plates were incubated at 37 °C for 3 h and fluorescence was measured at 590 nm after excitation at 560 nm using a Synergy HT microplate reader (BioTek). EC50 and IC50 values were calculated, when possible, fitting the data to a non-linear, sigmoidal regression model using GraphPad Prism 7.00 software.

*Effect on GAGs deposition by chondrocytes*. To test the effect of MTX-complexes on GAGs deposition by chondrocytes, HC-a cells were seeded on Nunclon^TM^ delta surface p60 plates (Thermo Scientific) at a density of 2·10^5^ cells/plate (n = 3) and cultured in complete chondrocyte medium containing 30 or 140 μM of the assayed compounds, depending on their cytotoxicity in this cell type. After 14 days of culture, media were removed and the plates were washed thrice with PBS. Next, 1 mL of a solution containing papain was added to the plates and incubated at 60 °C for 24 h to fully solubilize GAGs. Finally, GAGs content was quantified by the dimethyl methylene blue colorimetric assay (n = 6), measuring absorbance at 525 nm (filter 530 nm) in the microplate reader. GAGs content of each sample was normalized attending to the double-stranded DNA content (dsDNA). Quant-iT™ PicoGreen™ dsDNA assay kit was used for DNA quantification, measuring fluorescence at 520 nm after exciting at 480 nm.

*Effect on NO production by LPS-stimulated macrophages*. Additional anti-inflammatory properties of MTX complexes were assessed by analyzing the NO release by basal and LPS-stimulated RAW 264.7 macrophages treated with the synthesized compounds. Briefly, cells were seeded in sterile 96-well culture plates at a density of 2 × 10^5^ cells/mL and incubated for 24 h until confluence. Next, cells were incubated for 24 h with different concentrations of MTX derivatives (n = 6). After this, supernatants were collected and mixed in a ratio 1:1 *v/v* with the Griess reagent kit for nitrite determination. The absorbance at 540 nm was measured after incubation for 15 min at room temperature in dark and results were interpolated in a calibration curve constructed with known concentrations of sodium nitrite (NaNO_2_). In parallel, cells were incubated for 3 h with 100 μL of alamarBlue^®^ solution as described above for assessing their mitochondrial activity by measuring fluorescence at 590 nm after excitation at 560 nm.

*Data analysis and Statistics*. Results were expressed as means ± standard deviations of at least two experiments carried out with multiple biological replicates. Exact replicates are indicated in each biological experiment. Statistical analyses were performed using OriginPro 8 (version 8.0274) and GraphPad Prism 7.00 software. One-way ANOVA followed by the Tukey post hoc test was used to evaluate differences among groups, considering *p* ˂ 0.05 as statistically significant.

### 4.4. Encapsulation of MTX Derivatives into Chitosan Microparticles

*Preparation of loaded chitosan microparticles*. MTX- and MTX-complexes-loaded chitosan microparticles were synthesized by spray-drying. Briefly, first 125 mg of chitosan was dissolved in 25 mL of acetic acid 1% at gentle agitation at room temperature overnight. Next, 10 mL of aqueous solutions containing the different drugs at 1 mg/mL for MTX, SrMTX, and MgMTX (final drug content of 7.5% (*w*/*w*)), or 0.5 mg/mL for ZnMTX (final drug content of 3.75% (*w*/*w*)), was added dropwise to the corresponding chitosan solution under magnetic stirring. After homogenization, solutions were spray-dried using a mini spray dryer B-290 (Büchi Labortechnik AG, Flawil, Switzerland). The conditions selected for the process were aspirator efficiency of 80%, pump power of 20%, and inlet air temperature of 140 °C.

*Characterization of microparticles*. Particle sizes were characterized by mean diameter (D_10_, D_50_, D_90_), and span was calculated as Span = [(D_90_ − D_10_)/D_50_], using a Laser Diffraction Particle Size Analyser Coulter LS230 (Beckman, Electronics, USA) connected to a small volume module plus, and distributions were obtained using a software LS32. Around 20 mg of each sample was dispersed in 2 mL of distilled water and three measurements were recorded. Additionally, the morphology of the different microparticles was examined using a XL30 ESEM scanning electron microscope (Phillips). Samples were directly mounted on stubs and coated with a gold-palladium alloy using a sputter coater Polaron SC7640 (Thermo Scientific).

*Determination of the drug content in the microparticles*. Drug content and encapsulation efficiencies (EE) were calculated according to a previously described methodology [49]. Briefly, known masses of the different loaded microparticles were dispersed in 40 mM NaOH and the drugs were extracted for 48 h under magnetic stirring. Next, dispersions were passed through 13-mm polytetrafluoroethylene (PTFE) syringe filters (0.2 μm) and the different drug concentrations were determined by spectrophotometric techniques, measuring the absorbance of the samples at 304 nm with a UV-VIS NanoDrop OneC spectrophotometer (Thermo Scientific). The EE was calculated from the ratio of the actual drug content to the theoretical one and expressed as a percentage.

*In vitro drug release studies*. Known masses of the different loaded microparticles were placed over 20 μm Transwell^®^ culture plate inserts (Corning) and they were introduced in a 24-well culture plate containing 1 mL/well of PBS pH 7.4 at 37 °C to mimic physiological conditions. Aliquots of 5 μL were taken at different times and absorbance at 304 nm was measured using the UV-VIS NanoDrop OneC Spectrophotometer. Obtained data were plotted as release percentages.

## 5. Conclusions

In this article, we have reported the synthesis and characterization of metallic MTX derivatives bearing divalent cations such as Sr^2+^, Zn^2+^, and Mg^2+^ with novel molecular formula. This is the first time that metal-containing MTX derivatives are studied in the context of an autoimmune disease like RA, and it encompasses their most in-depth biological characterization so far. These complexations increased the thermal stability of the commercial drug and added new biological properties without affecting the intrinsic immunosuppressant properties of MTX. Therefore, these new derivatives should be studied more in depth using in vivo models, as an alternative to the gold standard treatment in cases of RA, where they could increase GAGs deposition and reduce the NO production, facilitating the regeneration of the damaged tissues. Also, their effects on periarticular bone cells, where bone erosions are found in RA, could be addressed in future in vitro and in vivo studies, due to the well-described regenerative effects of the selected cations. Lastly, as a proof of concept, we have successfully loaded these new compounds into polymeric microparticles, showing that they can be encapsulated and released in situ, and that, by using more complex delivery systems, a better controlled release could be achieved at the site of application in RA-affected joints.

## Figures and Tables

**Figure 1 ijms-23-10054-f001:**
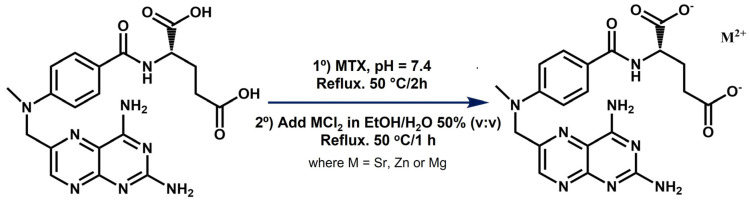
Scheme for the synthesis of SrMTX, ZnMTX, and MgMTX complexes.

**Figure 2 ijms-23-10054-f002:**
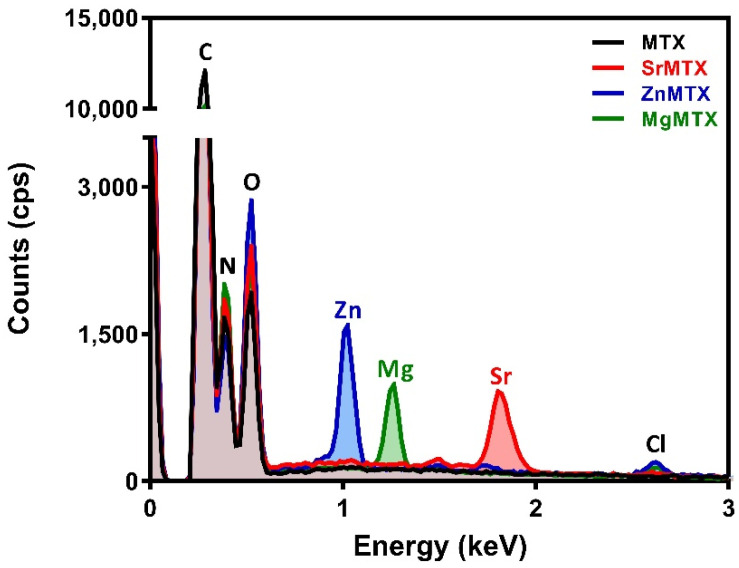
EDAX spectra registered for MTX and its derivatives.

**Figure 3 ijms-23-10054-f003:**
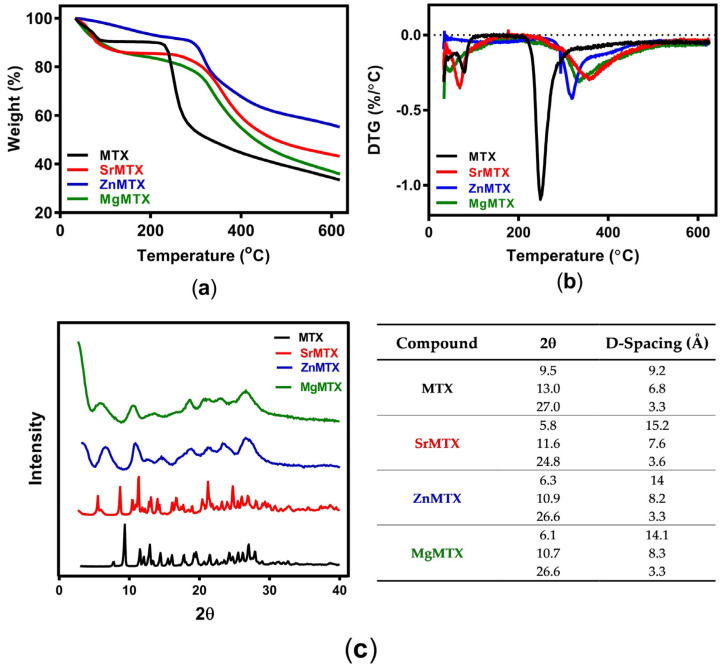
Physicochemical characterization of MTX and its derivatives: (**a**) TGA analysis; (**b**) DTG; (**c**) XRD analysis and main reflection peaks with calculated D-spacing.

**Figure 4 ijms-23-10054-f004:**
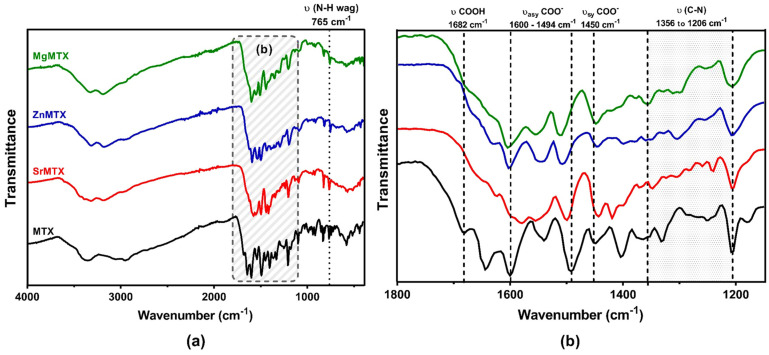
ATR–FTIR spectra of MTX and its derivatives (**a**) and zoomed-in region for their fingerprint determination (**b**). The most relevant vibrational frequencies of MTX are highlighted.

**Figure 5 ijms-23-10054-f005:**
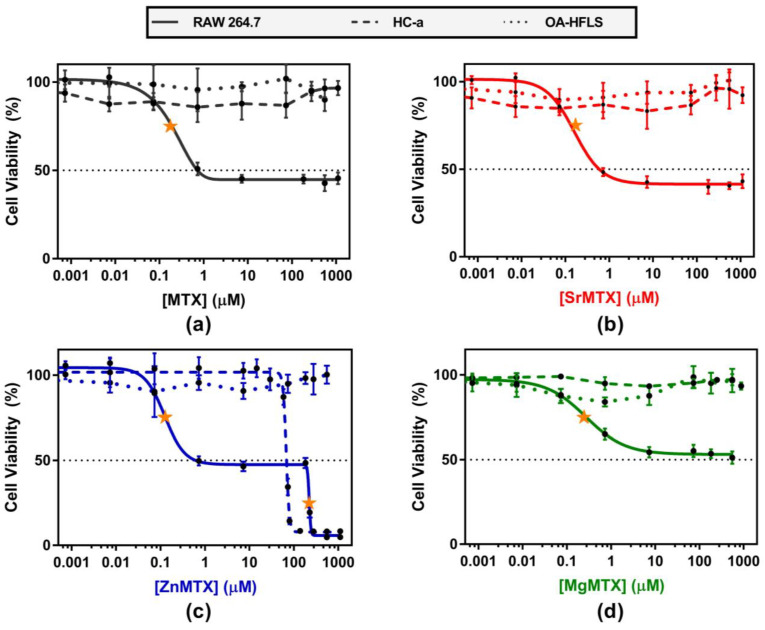
Cytotoxicity of (**a**) MTX; (**b**) SrMTX; (**c**) ZnMTX; (**d**) MgMTX on cultures of RAW 264.7 macrophages, HC-a cells, and OA-HFLS cells. EC50 values, calculated by non-linear regression analyses, are pointed out with a yellow star.

**Figure 6 ijms-23-10054-f006:**
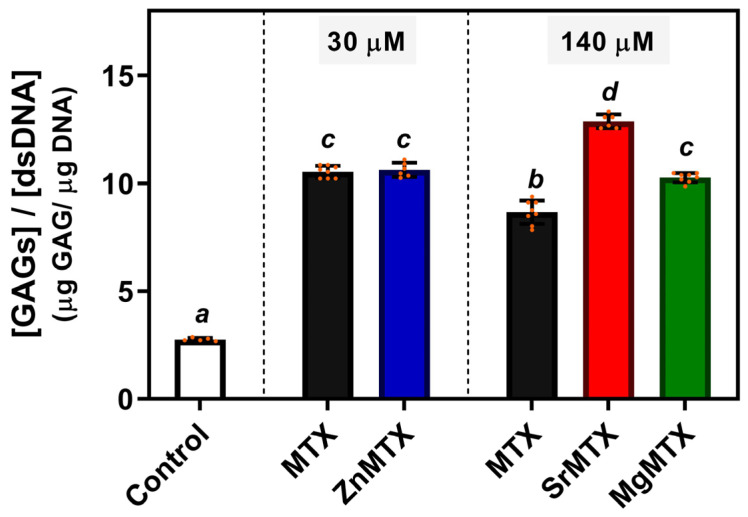
GAGs deposited by HC-a cells after 14 days of culture. Mean values, individual values (orange dots), standard deviation (n = 6), and one-way ANOVA results comparing among conditions are included. Different letters denote significant differences (*p* < 0.05, Tuckey post hoc test).

**Figure 7 ijms-23-10054-f007:**
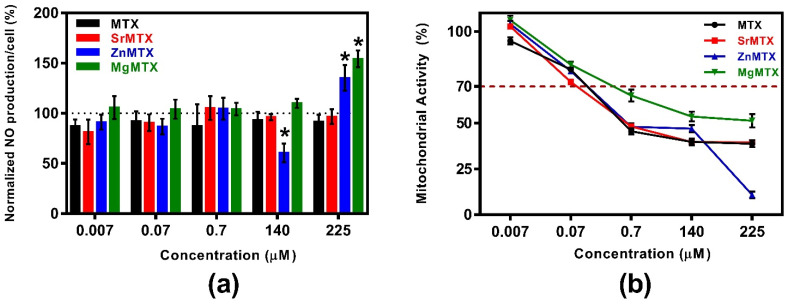
Additional anti-inflammatory properties of MTX and its derivatives: (**a**) Ratio of normalized NO production divided by the mitochondrial activity of non-LPS-stimulated RAW264.7 macrophage cultures; (**b**) Mitochondrial activity of non-LPS-stimulated RAW264.7 macrophage cultures. Mean values, standard deviation (n = 8), and the one-way ANOVA results comparing each condition with control cultures stimulated with LPS (100%) at a significance level of * *p* < 0.05 are included.

**Figure 8 ijms-23-10054-f008:**
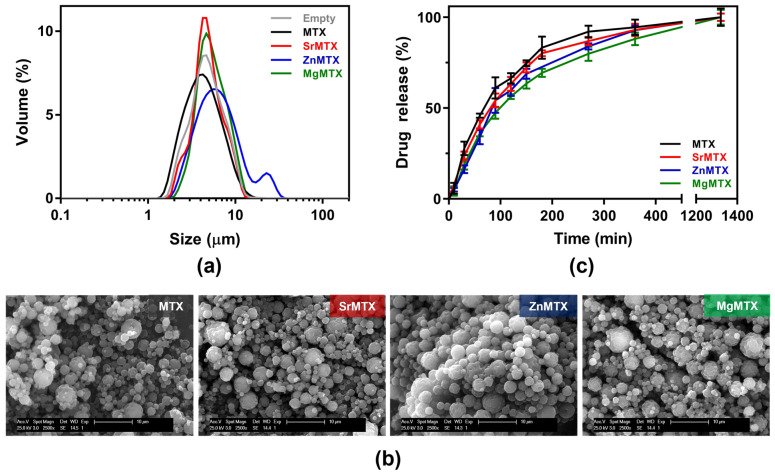
Characterization of chitosan microparticles loaded with MTX and its derivatives: (**a**) Size dispersion analysis, (**b**) Scanning electron microscopy images, and (**c**) Cumulative drug-release profiles.

**Table 1 ijms-23-10054-t001:** Molecular parameters and molecular formulas of metal-MTX complexes.

MTX Derivative	Molecular Formula	Mw (Da)	Metal *^a^*		Coordinated Water *^b^*	
			Cal %	Found %	Cal %	Found %
SrMTX	SrC_20_H_22_N_8_O_5_ 5H_2_O	632.12	13.86	16.06	14.24	14.27
ZnMTX	ZnC_20_H_22_N_8_O_5_ 1H_2_O	537.83	12.15	16.29	2.76	3.04
MgMTX	MgC_20_H_22_N_8_O_5_ 4H_2_O	550.79	4.83	6.91	12.94	13.01

*^a^* Determined by ICP spectroscopy. *^b^* Determined by TGA.

**Table 2 ijms-23-10054-t002:** Characterization of chitosan microparticles loaded with MTX and its derivatives.

Drug	Yield (%)	Mean Diameter (μm)	Size Span	Drug Content (% *w*/*w*)	Encapsulation Efficiency (%)	Loaded Drug Content (% *w*/*w*)
MTX	34.44	7.57	1.40	8	69.76 ± 1.40	5.58
SrMTX	51.20	5.96	1.12	8	93.09 ± 3.37	7.45
ZnMTX	42.23	11.98	1.80	4	60.98 ± 2.35	2.44
MgMTX	42.08	6.34	1.06	8	94.24 ± 3.83	7.54

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
