# Peer review of "Development of Methotrexate Complexes Endowed with New Biological Properties Envisioned for Musculoskeletal Regeneration in Rheumatoid Arthritis Environments"

_ijms, 2022, doi:10.3390/ijms231710054_

Round 1

Reviewer 1 Report

In this original research article, the authors synthesized and characterized methotrexate complexed with various metals for possible applications in rheumatoid arthritis patients. The topic is relevant and interesting, good and extensive results part. However, there are several issues in the manuscript which should be improved and clarified before publication.

1.     Abbreviations are explained when they first appear in the main text, even if they have been included in the abstract, and contribute to making the text easier to read and the information conveyed more efficient. Furthermore, there are repeating structures for which abbreviations may be used (e.g., RA for rheumatoid arthritis). According to the Instructions for authorsAcronyms/Abbreviations/Initialisms should be defined the first time they appear in each of three sections: the abstract; the main text; under the first figure or tablewhen defined for the first time, the acronym/abbreviation/initialism should be added in parentheses after the written-out form.” Revise the entire manuscript in this regard.

2.     The possibility of developing MTX for biomedical applications requires a more detailed description of the current safety profile. I suggest checking and referring to the following topical work https://doi.org/10.3390/cells10112857 that will help you to improve the information provided(

3.     For a comprehensive approach to the topic, as the last paragraph of Discussion section, it is advisable to present the main limitations and strengths of your study.

Reviewer 2 Report

The authors described a protocol to prepare metal-MTX complexes in high yields (>90%), and encapsulated the complexes with chitosan to form microparticles for in situ delivery in rheumatoid-arthritis-affected joints. A couple of major concerns need be addressed: (1) The authors should discuss the discrepancies of metal contents (Cal % versus Found %) in Table 1, and characterize the metal-MTX complexes by additional techniques (e.g., NMR, X-ray crystallographic analysis) to identify the binding sites. Multiple binding sites are possible. (2) Half-life values of drug release systems should be provided and discussed. Although no burst release is observed with the drug-loaded microparticles, it seems greater than 50% drug is released within 100 min. Does such fast release kinetics satisfy the requirement of in situ drug delivery and release? If not, additional experiments to optimize particle formulation need be performed.

Reviewer 3 Report

The topic of methotrexate use in rheumatoid arthritis and other autoimmune disorders is still of high interest even with new biological and targeted associated therapies. The aspect of tissue regeneration is poorly understood and even poorly studied, and till now we don't have any reliable proof of it. regarding this aspect row 43 at row 43 it is mentioned "there are some repots on the healing..." but no reference is added. Regarding methodology the effect of different forms of methotrexate was studied on cartilage and synovial cells, but not on periarticular bone cells (the aria where bone erosions are found in rheumatoid arthritis). Also, since we already now that the in vivo effect of methotrexate on rheumatological patients is seen after at least 8 weeks and fully consolidated at 6 months, it is not clear for me how long did the described experiment lasted.  Nevertheless the research idea is interesting, but further extesive research need to be done.

Round 2

Reviewer 2 Report

I don’t think the authors properly addressed the question regarding the  structural characterization of metal-MTX complexes. Is there any challenge to perform NMR measurements? It is necessary to discuss about the binding sites on the basis of sufficient data (e.g., at least NMR, if X-ray crystallization analysis is not available).

Author Response

We do appreciate the suggestion of the reviewer to perform a more thoughtful characterization of the newly synthesized drugs to look for additional binding sites. We have now updated the manuscript with new information addressing this issue. Please, see the attachment with all the details about the new information added to the manuscript. The changes in the manuscript are highlighted in red to facilitate the review.

Round 3

Reviewer 2 Report

The authors has made convincing NMR characterizations to support the conclusion regarding metal binding site(s). Questions have been resolved.